

# Allometry of sexual size dimorphism in turtles: a comparison of mass and length data

Koy W. Regis and Jesse M. Meik

Department of Biological Sciences, Tarleton State University, Stephenville, TX, United States

## ABSTRACT

**Background**. The macroevolutionary pattern of Rensch's Rule (positive allometry of sexual size dimorphism) has had mixed support in turtles. Using the largest carapace length dataset and only large-scale body mass dataset assembled for this group, we determine (a) whether turtles conform to Rensch's Rule at the order, suborder, and family levels, and (b) whether inferences regarding allometry of sexual size dimorphism differ based on choice of body size metric used for analyses.

**Methods**. We compiled databases of mean body mass and carapace length for males and females for as many populations and species of turtles as possible. We then determined scaling relationships between males and females for average body mass and straight carapace length using traditional and phylogenetic comparative methods. We also used regression analyses to evalutate sex-specific differences in the variance explained by carapace length on body mass.

**Results**. Using traditional (non-phylogenetic) analyses, body mass supports Rensch's Rule, whereas straight carapace length supports isometry. Using phylogenetic independent contrasts, both body mass and straight carapace length support Rensch's Rule with strong congruence between metrics. At the family level, support for Rensch's Rule is more frequent when mass is used and in phylogenetic comparative analyses. Turtles do not differ in slopes of sex-specific mass-to-length regressions and more variance in body size within each sex is explained by mass than by carapace length.

**Discussion**. Turtles display Rensch's Rule overall and within families of Cryptodires, but not within Pleurodire families. Mass and length are strongly congruent with respect to Rensch's Rule across turtles, and discrepancies are observed mostly at the family level (the level where Rensch's Rule is most often evaluated). At macroevolutionary scales, the purported advantages of length measurements over weight are not supported in turtles.

Corresponding author
Jesse M. Meik, meik@tarleton.edu

## INTRODUCTION

Body size is among the most frequently used variables in large-scale macroecological and evolutionary studies because it is a fundamental property of organisms relevant to physiology, ecology, anatomy, extinction risk, and genomic architecture (*Peters, 1986*; *Calder III, 1996*; *Cardillo et al., 2005*; *Lynch, 2007*). However, despite its transcendent

importance to nearly every aspect of biology, exactly what body size is remains unclear. Researchers often infer body size as a linear measurement based on anatomical landmarks, for example wing chord length in birds (*Gosler et al., 1998*); snout–vent length in most amphibians and reptiles, total length in others (*Olalla-Tárraga & Rodríguez, 2007*; *Meiri, 2010*), and various other linear measurements are often used among invertebrates. To other researchers, body size is equated with some other physical property such as mass or volume. More recently, advances in geometric morphometrics have allowed three-dimensional assessments of size and shape of biological structures (*Chiari et al., 2008*). The breadth of measurements used for body size brings the question of whether body size should be perceived as a real attribute, for which each of these variables might capture some aspects, or simply as an abstraction for which any of these subordinate concepts are interchangeable. On one hand, the frequent use of words such as "surrogate" or "proxy" in reference to the relationship between these measurements and body size implies that body size is perceived as a real latent variable. On the other hand, a widespread lack of interest or discussion about how well each of these variables might correspond to "true" body size implies an abstract conception of body size. Likewise, the frequent practice of combining different measurements of body size for the same analyses implies that at least empirically body size is often perceived as an abstract variable, with the justification being that the suites of variables used to infer body size are strongly correlated. However, strong correlations between variables do not necessarily ensure that they will yield the same statistical or biological inferences when used as proxies for each other. For example, the body size frequency distribution for a sample of squamate reptiles was found to be bimodal when length was used but was unimodal in a larger sample of length-derived mass estimates (*Cox, Butler & John-Adler, 2007*; *Feldman et al., 2016*).

Assuming that body size is a real attribute that is universal across life, to what extent does each of these surrogate measurements approximate this property? Sometimes an argument is presented to justify a particular measurement over others, but ultimately the proxies for body size most often selected seem to be based on convention or convenience (*Houle et al., 2011*; and e.g., *Iverson, 1982*; *Gibbons & Lovich, 1990*). Most types of linear measurements are of limited value in that they often are restricted to single taxa. Furthermore, each length measurement reflects selection on body size through allometric growth, and each anatomical feature may be subject to its own selective pressures and constraints, independent of or in conjunction with overall body size (*Andersson, 1994*).

If the underlying causes of body size relate to energetics of metabolism (i.e., metabolic theory; *Gilbooly et al., 2001*; *Brown et al., 2004*), then mass might hold unique status as a proxy for body size because it will more accurately reflect the amount of matter (i.e., number of cells) that must be organized and maintained by living systems. Mass also can be measured and directly compared across all clades, which makes it a particularly desirable variable for macroecological studies. These may be reasons why researchers often interpolate mass datasets from regression equations based on limited mass and length data, which we hereafter refer to as length-derived mass data (LDM data; *Pough, 1980*; *Iverson, 1982*; *Meiri, 2010*; *Burbrink & Myers, 2013*). Although mass data are often collected for amphibians

and non-avian reptiles, snout–vent length (or carapace length for turtles) is far more frequently reported and equated with body size. Mass data are often assumed to be more prone to fluctuations in body condition, digestive state, health, and reproductive status (*Stamps, 1983*; *Cox, Butler & John-Adler, 2007*; *Bonnet et al., 2010*), and these assumptions are frequently supported with intraspecific datasets (e.g., *Jacobson et al., 1993*; *Nagy et al., 2002*; *Stevenson & Woods Jr, 2006*). However, there is little empirical information addressing the relative variability of mass and length data at macro-scales, especially for non-avian reptiles, as most researchers focus on obtaining only one measurement of body size for macroevolutionary datasets. Moreover, the variability of a measurement does not necessarily indicate its appropriateness as a surrogate for body size. Biological conditions such as reproductive condition or starvation could be important (albeit transient) aspects of body size, at least physiologically.

As an empirical test of the congruency of mass and length body size data, we here evaluate inferences from each of these size metrics as they relate to the macroevolutionary pattern of Rensch's Rule (*Rensch, 1950*) in turtles (clade Chelonia). Rensch's Rule (RR) describes an allometric scaling relationship in body size where among closely related species, the size difference between males and females increases with overall body size in species where the males are the larger sex. When females are the larger sex, RR predicts sexual size dimorphism (SSD) to diminish as overall species body size increases.

Rensch's Rule has been widely studied in diverse taxa such as plants, arthropods, reptiles (including birds), and mammals (reviewed in *Fairbairn, 1997*). Support for the pattern is varied but is most often reported in male size-biased taxa, where sexual selection on males for larger size combined with genomic covariation between the sexes is assumed to result in the pattern (*Fairbairn & Preziosi, 1994*). This hypothesis also implies that large body size may result generally from male-biased size dimorphism. However, support for the pattern is less often found in female size-biased lineages (which comprise the majority of sexually dimorphic lineages), and the relative contributions of sexual selection and fecundity selection on body size in female size-biased lineages are less resolved (*Webb & Freckleton, 2007*). If a lineage displays the converse of RR, with strongly female size-biased species at larger overall sizes, then fecundity selection (perhaps together with antagonistic sexual selection for smaller males that could partition more of their energy budgets into mate-searching rather than growth) is often assumed to play a major role in SSD and body size in that group.

Studies of RR have used many different measurements of body size, usually whichever proxy of body size is most readily available for the taxon in question, and by doing so potentially equate disparate measurements as body size. Which type of measurement, and therefore implicit definition of body size, to use for analyses of RR has been a recurring issue in studies of SSD (*Lovich, Ernst & McBreen, 1990*; *Fairbairn, Blanckenhorn & Székely, 2007*). For example, *Székely, Lislevand & Figuerola (2007)* examined sexual dimorphism in body mass, wing length, tarsus length, bill length, and tail length in birds and found only weak correlations between these traits with respect to SSD. Despite these concerns, few studies have directly compared length measurements with mass data to determine  whether

they would yield different inferences relating to body size dimorphism (but see *Rising & Somers, 1989*; *Gibbons & Lovich, 1990*). The uncertainty of how to define body size as it relates to RR is as old as the hypothesis, because in Rensch's seminal work (*1950*), both length and mass data were combined in the same analyses to formulate RR.

Turtles are a model clade for which to study body size and SSD, as both attributes vary dramatically across lineages. Turtles include about 330 extant species, and individual species range in size from the diminutive speckled tortoise, *Homopus signatus* (Gmelin, 1789), attaining weights of ∼130 g and carapace lengths up to 9 cm (*Loehr, 2001*), to the leatherback sea turtle, *Dermochelys coriacea* (Vandelli, 1761), one of the largest extant reptiles, attaining weights of at least 650 kg and curved carapace lengths of over 2.1 m (*McClain et al., 2015*). The standard body size measurement in turtles is the taxon-specific straight carapace length (SCL), a linear measurement of the dorsal shell. When using carapace length, size comparisons between turtles and other groups are possible only through LDM data based on allometric regression equations of limited value (e.g., *Pough, 1980*). Carapace length often is lauded as a stable measurement of size across turtles, with little or no apparent seasonal or daily variation. However, species vary, especially by family, in the relative size and shape of the shell when compared with other aspects of body size. Furthermore, there is often ambiguity in what is actually being measured with carapace length. Curved carapace length (the non-Euclidian-distance over the curve of the shell) is sometimes substituted for SCL in larger taxa (e.g., sea turtles) for practicality, but these measurements cannot be directly compared to straight carapace length, owing to interspecific differences in shell shape. There are at least three different methodologies for measuring straight carapace length in turtles (and two for curved carapace length) that differ with respect to the anatomical start and end points (*Pritchard et al., 1983*; *Bolten, 1999*), and researchers may neglect to disclose what they mean by "straight carapace length," or, even worse, "carapace length." In contrast, body mass is less ambiguous (although still subject to error) and is seemingly widely recorded, but unfortunately as with other non-avian reptiles, mass is rarely reported in turtles (*Iverson, 1982*).

Rensch's Rule within turtles has had mixed support from previous studies. Early studies were limited in their taxonomic and geographic scope and did not employ phylogenetic comparative methods (*Berry & Shine, 1980*; *Iverson, 1985*; *Gibbons & Lovich, 1990*). Those studies found support for RR in Kinosternidae, but not in turtles overall. More recent studies have varied in phylogenetic hypotheses, body size metrics (i.e., mean versus maximum length), and types of regression analyses, and have found support for either RR or for isometry (*Cox, Butler & John-Adler, 2007*; *Stephens & Wiens, 2009*; *Ceballos, Hernández & Valenzuela, 2014*; *Halámková, Schulte & Langen, 2013*; *Werner et al., 2016*). At the family level, the Kinosternidae has continued to attract attention, as different studies arrived at different conclusions (*Ceballos & Iverson, 2014*). Most turtle species are female-biased in carapace length, but the RR pattern has been found most often in male size-biased taxa, so turtles are unusual in this respect (*Ceballos, Hernández & Valenzuela, 2014*). Despite the overall trend of female size-bias across turtles, the directionality of size dimorphism varies within most families, sometimes even within genera and species (*Lovich et al., 2010*). A recent study of spur-thighed tortoises (*Testudo graeca*) found that SSD was male-biased

in populations with large body size and female-biased in populations with small body size, showing an intraspecific pattern consistent with RR (*Werner et al., 2016*). Although a variety of methodologies have been employed, all previous studies of SSD in turtles have used only carapace length as a size metric, and therefore we evaluate, for the first time, whether the choice of body size measurement (length or mass) affects inferences about RR in turtles.

## MATERIALS & METHODS

### Data Collection

Data were compiled on average adult body mass and straight carapace length (SCL) for males and females for as many turtle species and populations as possible from primary literature, government agency reports, dissertations and theses, as well as unpublished sources. Body mass data came from 198 sources (Appendix S1). Data on SCL came mostly from the datasets of *Ceballos, Hernández & Valenzuela (2014)* and *Halámková, Schulte & Langen (2013)*, augmented with 122 sources that recorded both body mass and SCL (Appendix S1). For both datasets, data from captive populations were included to increase species coverage, but data on juveniles and gravid females were excluded when possible. Although captive turtles often exhibit different patterns of growth than do wild populations, common garden experiments indicate that the direction and magnitude of SSD are mostly consistent between captive and wild populations (*Ceballos & Valenzuela, 2011*; *Ceballos, Hernández & Valenzuela, 2014*). Moreover, only seven of 146 species used for analyses of mass and four of 241 populations used for analyses of SCL were from captive populations; therefore, any influence of captivity would likely be minimal at the scale of our study. When only ranges of values were reported, the midpoint was calculated and used as average body size for both SCL and mass. In a few instances, maximum values were used if no averages were reported. Inferences from previous studies have been mixed as to whether maximum values yield similar results to mean values (e.g., *Fitch, 1981*; *Lovich & Gibbons, 1992*; *Boback & Guyer, 2003*); however, once again maximum values constituted a small portion of our dataset (nine of 146 species for mass and 19 of 241 species for SCL). Two morphotypes (i.e., "saddlebacked" and "domed") of the *Chelonoidis* species complex of Galapagos tortoises were included in non-phylogenetic analyses, as populations of *Chelonoidis nigra* (Quoy & Gaimard, 1824); we chose to retain this conservative taxonomy as recent nomenclatural changes for Galapagos tortoises have not yet stabilized. For three species (*Apalone ferox*, *Kinosternum integrum*, and *Pseudemys gorzugi*), we combined data on body mass of males and females from different sources, because no single study reporting body mass for both sexes could be found. Prior to all analyses, body mass and SCL were log-transformed. When we had data on more than one population of a species for both datasets, we randomly selected one population per species for analysis. However, when indicated, we also performed analyses using the full population-level datasets (i.e., multiple populations of some species).

As a preliminary assessment of trends in SSD across families, we calculated the commonly used index of *Lovich & Gibbons (1992)* for each species and present the means of these

indices. This index is calculated as (larger sex/smaller sex) +1 if males are larger and −1 if females are larger, arbitrarily set as positive if females are larger and negative if males are larger. The index is symmetric around zero and comparable to a percent difference in size. We calculated the dimorphism index using log-transformed SCL and mass data. We present the mean of the ratios, not the ratio of the means, in order to illustrate uniformity or variability in directionality of dimorphism, and because the ratio of the means is overly influenced by the largest species.

## Analysis of Rensch's rule

Rensch's Rule typically is analyzed by regressing log-transformed male body size against log-transformed female body size (*Fairbairn, 1997*; *Ceballos, Hernández & Valenzuela, 2014*; *Halámková, Schulte & Langen, 2013*). When log-female size is plotted on the *x*-axis and log-male size is plotted on the *y*-axis, positive allometry (a slope greater than one) represents a pattern consistent with RR, and negative allometry (a slope less than one) represents the converse of RR. A slope not significantly different from one represents isometry. Standardized major axis (SMA) regression was selected over ordinary least squares regression, as there is no *a priori* reason to suspect differences in measurement error between the sexes. We performed SMA regression using the package "smatr" (*Warton et al., 2012*) in R software version 3.2.0 (*R Core Team, 2015*). The 95% confidence intervals (CI) of the regression slopes were calculated, with CI lower limit >1 indicating RR, CI upper limit <1 indicating the converse of RR, and a CI range that includes 1 indicating isometry. Rensch's Rule analyses were performed across all turtles and also performed at the suborder level (i.e., Cryptodira and Pleurodira) and at the family level (for families with data available for seven or more species).

To account for the differences in shared phylogenetic history among species, and as is standard for analyses of RR, we repeated our analyses using phylogenetic comparative methods after first testing the datasets for phylogenetic signal with Blomberg's *K* statistic (*Blomberg, Garland & Ives, 2003*) using the R package "picante" (*Kembel et al., 2010*). A significant *K*-value indicates that a particular tree explains more variance than a star phylogeny, and therefore that phylogenetic comparative methods would be justified (i.e., non-independence of trait values in closely related species might influence regression parameters). A *K*-value of more than or less than 1 indicates that species resemble their relatives in that trait more or less, respectively, than would be expected under a Brownian motion model of evolution. Male body mass, female body mass, male SCL, and female SCL were each evaluated separately for *K*. The phylogeny used was an unpruned version of the turtle supertree from *Angielczyk, Burroughs & Feldman (2015)* provided by the authors. When polytomies occurred, they were broken into random dichotomies. We were able to match 145 species with body mass data and 241 species with carapace length data to the phylogeny. These data were transformed using phylogenetic independent contrasts (PIC) with the "ape" package (*Paradis, Claude & Strimmer, 2004*) in R. The PICs were then regressed using SMA regression, with the intercept fitted through the origin, as recommended by *Garland, Harvey & Ives (1992)*. As with the non-phylogenetic regressions,

the estimated slopes and their 95% confidence intervals were used to assess conformity to RR, its converse, or isometry.

## Mass-carapace length relationships

It is plausible, due to sexual selection on males or the metabolic and gestational constraints on reproduction in females, that growth could be partitioned differently between mass and length within sexes. Furthermore, if gravidity in females cannot reliably be detected by researchers in the field, females should show a more variable relationship between mass and length than males, which might add noise to macroevolutionary analyses. To determine if males and females differed in their relationships between body mass and SCL, their log-transformed SCLs were regressed against their log-transformed body masses, and then a Chow test (*Chow, 1960*) was performed to test for differences in the coefficients of determination of regressions of males and females combined and of males and females separately. We performed these analyses on all 208 populations for which data on male and female mass and SCL were available. We chose not to use phylogenetic analysis for this question and to include multiple populations for some species, as our purpose was simply to assess differences in variance explained by body size metrics between sexes.

# RESULTS

## Data summary

Body mass data included 307 populations representing 146 of the approximately 330 turtle species (*Van Dijk et al., 2014*), and SCL data included 581 populations, representing 242 living species (Table 1; full dataset is included as Data S1). All fourteen turtle families were present in both datasets. Of the populations used for analysis of RR, males were heavier in 38 of 146 species (26.0%), females were heavier in 104 species (71.2%), and 4 species (2.7%) had negligible SSD (i.e., <2% difference). Similarly, males had longer carapaces in 66 of 242 species (27.3%), females had longer carapaces in 161 species (66.5%), and 15 species (6.2%) had negligible SSD. The direction of SSD generally was consistent at the family level between body mass and SCL, with the exception of families Pelomedusidae and Kinosternidae. As noted by *Lovich & Gibbons (1992)*, the choice of mass or length data influences the perceived magnitude of SSD, and we observed that indices calculated from body mass were often, but not always, more extreme, than indices calculated from SCL (Table 1).

## Rensch's rule across chelonia

Without phylogenetic comparative methods, body mass and carapace length disagreed slightly regarding patterns of allometry in turtles (only in terms of the statistical threshold indicating allometry). Body mass ($n = 146$, $r^2 = 0.858$, $P < 0.001$) showed positive allometry (RR; $b = 1.066$; 95% CI [1.002–1.135]; for $H_0$ $b = 1$, $P = 0.042$). Straight carapace length ($n = 241$, $r^2 = 0.789$, $P < 0.001$) showed a marginally non-significant positive allometry ($b = 1.057$; 95% CI [0.997–1.120]; for $H_0$ $b = 1$, $P = 0.064$), thus supporting isometry (Fig. 1).

Blomberg's $K$-values indicated that both body mass and SCL had moderate phylogenetic signal for each sex ($P$-values < 0.001). Male mass had the lowest $K$, at 0.545, female mass

Regis and Meik (2017), *PeerJ*, DOI 10.7717/peerj.2914

**Table 1  Summary of the datasets.** The data used for phylogenetic analyses (one population per species) are listed first, and the data for all populations (including multiple populations of some species) follows in parentheses. Mean dimorphism index is that of *Lovich & Gibbons (1992)*, with negative values indicating male size bias.

| Family | Extant species | Mass species | SCL species | Male Mean size | | Female Mean size | | Mean dimorphism index | |
|---|---|---|---|---|---|---|---|---|---|
| | | | | Mass (g) | SCL (cm) | Mass (g) | SCL (cm) | Mass | SCL |
| Carettochelyidae | 1 | 1 (1) | 1 (3) | 9,500 (9,500) | 45.4 (40.3) | 16,000 (16,000) | 52.3 (47.2) | 0.684 (0.684) | 0.152 (0.177) |
| Chelidae | 54 | 26 (36) | 39 (78) | 1,274 (1,404) | 22.8 (21.9) | 1,987 (2,114) | 25.2 (24.8) | 0.502 (0.484) | 0.118 (0.140) |
| Cheloniidae | 6 | 4 (6) | 5 (15) | 52,132 (67,255) | 77.7 (83.9) | 57,414 (74,683) | 81.6 (87.8) | 0.096 (0.107) | 0.052 (0.045) |
| Cheyldridae | 4 | 2 (10) | 3 (9) | 22,129 (19,396) | 44.7 (40.6) | 10,530 (10,103) | 39.1 (34.9) | −0.778 (−0.713) | −0.136 (−0.152) |
| Dermatemydidae | 1 | 1 (4) | 1 (2) | 6,396 (6,267) | 38.2 (40.7) | 6,622 (6,617) | 34.2 (41.2) | 0.035 (0.060) | −0.118 (−0.001) |
| Dermochelyidae | 1 | 1 (2) | 1 (2) | 32,0000 (41,3500) | 155 (161.8) | 38,7600 (39,7850) | 147.1 (161.5) | 0.211 (−0.016) | −0.054 (−0.005) |
| Emydidae | 49 | 20 (68) | 48 (129) | 736 (640) | 17.7 (17.1) | 1,754 (1,155) | 23.4 (21.8) | 1.99 (1.25) | 0.386 (0.315) |
| Geoemydidae | 72 | 29 (47) | 52 (106) | 1,487 (1,619) | 19.5 (19.6) | 2,835 (2,809) | 23.6 (24.1) | 1.61 (1.12) | 0.238 (0.261) |
| Kinosternidae | 24 | 12 (22) | 23 (59) | 344 (304) | 14.4 (13.6) | 315 (300) | 13.7 (13.0) | 0.059 (0.098) | −0.041 (−0.046) |
| Pelomedusidae | 19 | 3 (4) | 4 (7) | 2,550 (2,662) | 33.1 (32.8) | 3,607 (3,255) | 32.6 (32.3) | 0.637 (0.387) | −0.029 (−0.025) |
| Platysternidae | 1 | 1 (1) | 1 (2) | 367 (367) | 13.2 (16.7) | 306 (306) | 12.2 (15.1) | −0.201 (−0.201) | −0.080 (−0.098) |
| Podocnemididae | 8 | 7 (10) | 8 (20) | 2,615 (2,387) | 30.8 (29.3) | 4,289 (4,435) | 37.8 (37.4) | 0.925 (1.13) | 0.223 (0.278) |
| Testudinidae | 68 | 29 (81) | 43 (124) | 25,690 (15,809) | 25.9 (26.4) | 11,508 (7,986) | 25.2 (25.6) | 0.065 (0.192) | 0.062 (0.048) |
| Trionychidae | 27 | 10 (15) | 12 (25) | 19,507 (17,187) | 36.5 (37.1) | 28,373 (23,720) | 45.7 (46.9) | 1.45 (1.27) | 0.436 (0.413) |
| All Chelonians | 335 | 146 (307) | 241 (581) | 11,303 (10,381) | 24.0 (24.1) | 10,221 (8,858) | 26.9 (26.8) | 0.853 (0.650) | 0.182 (0.168) |

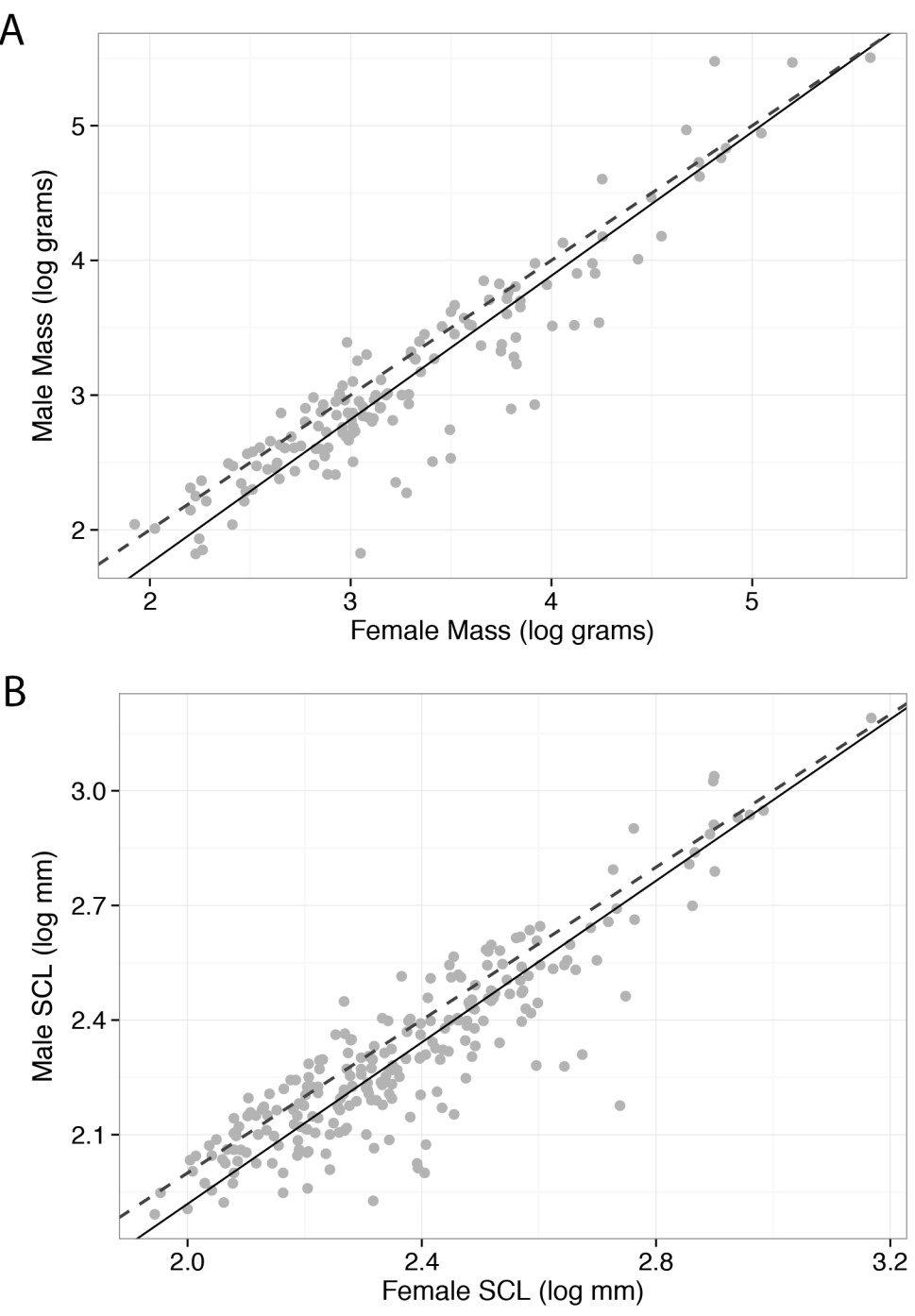

**Figure 1   Standardized major axis regression of male and female body size measurements of turtle species.** Body mass (A) supported Rensch's rule, and straight carapace length (B) supported an isometric relationship between body size and sexual size dimorphism. Dashed lines represent isometry.

was 0.604, female carapace length was 0.619, and male carapace length 0.636, suggesting that mass has slightly more phylogenetic signal. However, the similar levels of phylogenetic signal in both sexes provides support for strong genetic correlation in body size between the sexes (*Fairbairn, 1997*). In the phylogenetic comparative analyses, conformity to RR was supported by both body mass and SCL with high congruence of regression parameter estimates (Fig. 2). The SMA regression for PICs of body mass had a slope of 1.099 ($n = 145$; 95% CI [1.029–1.173]; $r^2 = 0.844$; for $H_0$ $b = 1$, $P = 0.0048$), and the PICs for SCL had a slope of 1.093 ($n = 241$; 95% CI [1.029–1.16]; $r^2 = 0.773$; for $H_0$ $b = 1$, $P = 0.0042$).

### Rensch's rule at suborder and family level

Despite the overall pattern of RR in turtles, positive allometry was not universal among turtle families. The choice of body size metric and whether phylogenetic comparative methods were employed both influenced inferences of SSD allometry in several turtle families. Using non-phylogenetic analyses, RR was not supported in the suborder Pleurodira or any of its families, suggesting the overall pattern of RR in turtles may result mostly from positive allometry in the Cryptodira suborder (Table 2). For body mass, we found support for RR in the suborder Cryptodira and in two of its families, Testudinidae and Trionychidae; however, using SCL we detected RR for only the family Testudinidae. All other non-phylogenetic regression slopes at the suborder and family level supported isometry (Table 2).

Rensch's rule was more often detected in a phylogenetic context at the family level, particularly for body mass (Table 3). Using body mass, support for Rensch's Rule was found in Geoemydidae and Kinosternidae, in addition to Testudinidae, Trionychidae, and Cryptodira overall. Using SCL, we found support for RR only in Testudinidae and Cryptodira overall. As with non-phylogenetic regressions, the remaining families demonstrated isometry.

### Mass-Carapace length relationships

As expected, both males and females showed strong relationships between SCL and body mass when including all populations for which we had obtained data for both variables (including multiple populations of several species) (Fig. 3). For both sexes combined, $r^2$ was 0.932 ($n = 416$; $b = 0.345$, $P < 0.0001$). Analyzed separately, $r^2$ was 0.910 for males ($n = 208$, $b = 0.338$) and 0.954 for females ($n = 208$, $b = 0.341$). Although females had slightly higher variance in SCL explained by body mass, neither the slopes ($P = 0.5$) nor the coefficients of determination (Chow test: $F = 2.176$, $P = 0.115$) were significantly different between the sexes.

## DISCUSSION

Our allometric analyses include the largest independent body mass and SCL datasets yet brought to bear on the question of RR in turtles. At the deepest phylogenetic scale, results show remarkable congruence in estimates of positive allometry for both body mass and SCL data (Fig. 2), and thus indicate that overall turtles follow RR. However, we noted some discrepancies in inferences of RR depending on analyses (non-phylogenetic vs. phylogenetic) and body size metric used, suggesting that these factors will partly influence

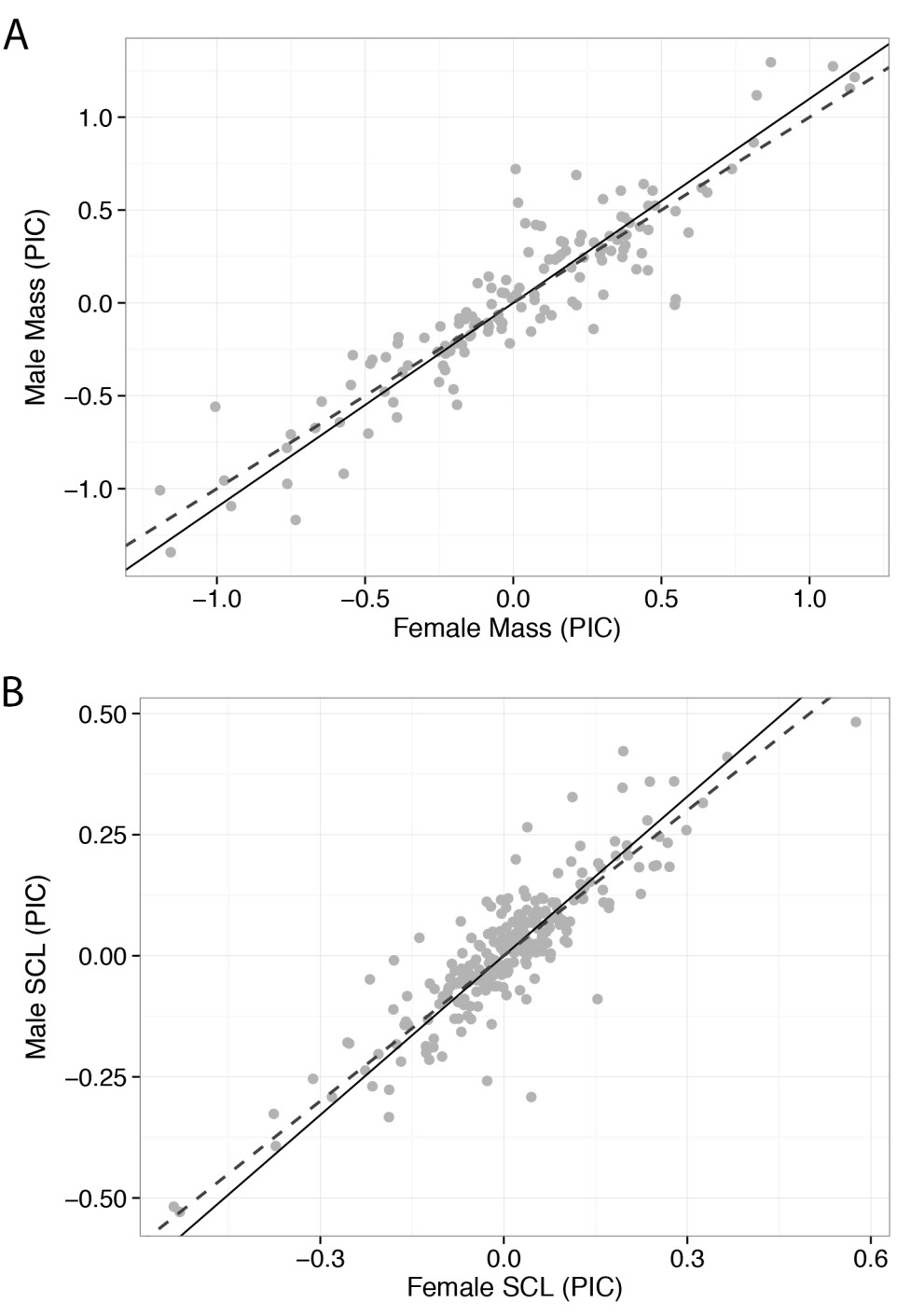

**Figure 2** **Phylogenetic independent contrasts regressed using standardized major axis regression for male and female body size measurements of turtle species.** Body mass (A) and straight carapace length (B) both supported Rensch's Rule. Dashed lines represent isometry, and lines of best fit are forced through the origin. At this scale, mass and length body size measurements yielded comparable results.

**Table 2** **Results of standardized major axis regressions at the suborder and family levels for both body mass and carapace length data without adjustment for phylogeny.** Em dashes represent an isometric relationship, and RR indicates Rensch's Rule. The families Pelomedusidae, Cheloniidae, and the 4 monotypic families were omitted from family analyses due to insufficient sample sizes, but are included in the analyses of turtle suborders (bold text).

| Clade | Body mass | | | | | Straight carapace length | | | | |
|---|---|---|---|---|---|---|---|---|---|---|
| | *n* | Intercept | Slope | 95% CI | Pattern | *n* | Intercept | Slope | 95% CI | Pattern |
| **Cryptodira** | 110 | −0.984 | 1.081 | 1.008, 1.158 | RR | 190 | −0.199 | 1.058 | 0.991, 1.129 | — |
| Emydidae | 20 | 0.472 | 0.826 | 0.563, 1.213 | — | 48 | −0.178 | 1.023 | 0.832, 1.256 | — |
| Geoemydidae | 29 | −0.702 | 1.016 | 0.795, 1.30 | — | 52 | −0.097 | 1.008 | 0.850, 1.194 | — |
| Kinosternidae | 12 | −1.278 | 1.227 | 0.866, 1.738 | — | 23 | −0.272 | 1.136 | 0.952, 1.356 | — |
| Testudinidae | 29 | −1.826 | 1.230 | 1.158, 1.306 | RR | 43 | −0.531 | 1.217 | 1.130, 1.311 | RR |
| Trionychidae | 10 | −4.560 | 1.400 | 1.059, 1.851 | RR | 12 | −0.292 | 1.068 | 0.705, 1.616 | — |
| **Pleurodira** | 36 | 0.169 | 0.926 | 0.783, 1.094 | — | 51 | −0.578 | 1.005 | 0.863, 1.170 | — |
| Chelidae | 26 | 0.274 | 0.916 | 0.738, 1.136 | — | 39 | −0.182 | 1.057 | 0.876, 1.275 | — |
| Podocnemididae | 7 | −0.961 | 1.050 | 0.595, 1.852 | — | 8 | 0.199 | 0.890 | 0.550, 1.442 | — |

**Table 3** **Results of standardized major axis regressions at the suborder (bold text) and family levels using phylogenetic independent contrasts.** Em dashes represent an isometric relationship and RR indicates Rensch's Rule.

| Clade | Body mass | | | | Straight carapace length | | | |
|---|---|---|---|---|---|---|---|---|
| | *n* | Slope | 95% CI | Pattern | *n* | Slope | 95% CI | Pattern |
| **Cryptodira** | 110 | 1.148 | 1.065, 1.237 | RR | 190 | 1.121 | 1.046, 1.202 | RR |
| Emydidae | 20 | 1.008 | 0.681, 1.493 | — | 48 | 1.024 | 0.832, 1.259 | — |
| Geoemydidae | 29 | 1.305 | 1.021, 1.666 | RR | 52 | 1.157 | 0.944, 1.418 | — |
| Kinosternidae | 12 | 1.495 | 1.103, 2.026 | RR | 23 | 1.102 | 0.943, 1.289 | — |
| Testudinidae | 29 | 1.212 | 1.137, 1.293 | RR | 43 | 1.235 | 1.118, 1366 | RR |
| Trionychidae | 10 | 1.602 | 1.215, 2.112 | RR | 12 | 1.190 | 0.766, 1.850 | — |
| **Pleurodira** | 36 | 0.861 | 0.709, 1.046 | — | 51 | 0.962 | 0.817, 1.133 | — |
| Chelidae | 26 | 0.855 | 0.679, 1.076 | — | 39 | 1.056 | 0.878, 1.271 | — |
| Podocnemididae | 7 | 1.023 | 0.594, 1.760 | — | 8 | 0.866 | 0.592, 1.267 | — |

the detection of patterns of allometry in SSD. Most discrepancies related to discerning a pattern of positive allometry from isometry at the family level, which is the level most often used in comparative studies of SSD (e.g., *Cox, Butler & John-Adler, 2007*; *Székely, Lislevand & Figuerola, 2007*; *Ceballos, Hernández & Valenzuela, 2014*). In contrast to some previous studies (e.g., *Lindeman , 2008*; *Ceballos et al., 2013*), we found no support for a pattern converse to RR in any chelonian clades. With the mass dataset, we were able to detect RR in four families, rather than just Testudinidae as with the SCL dataset, despite that sample sizes were larger across all families for the SCL dataset. This result might be expected given that volume scales to the cube of linear measurements (under the simplistic assumption of isometry), and therefore differences in SSD will generally be more extreme with mass than with length datasets, as it is in turtles (*Lovich & Gibbons, 1992*; Table 1). Although it would be tempting to conclude that SCL will give more conservative estimates of body size allometry, it is important to recognize that SCL and mass are different measures and

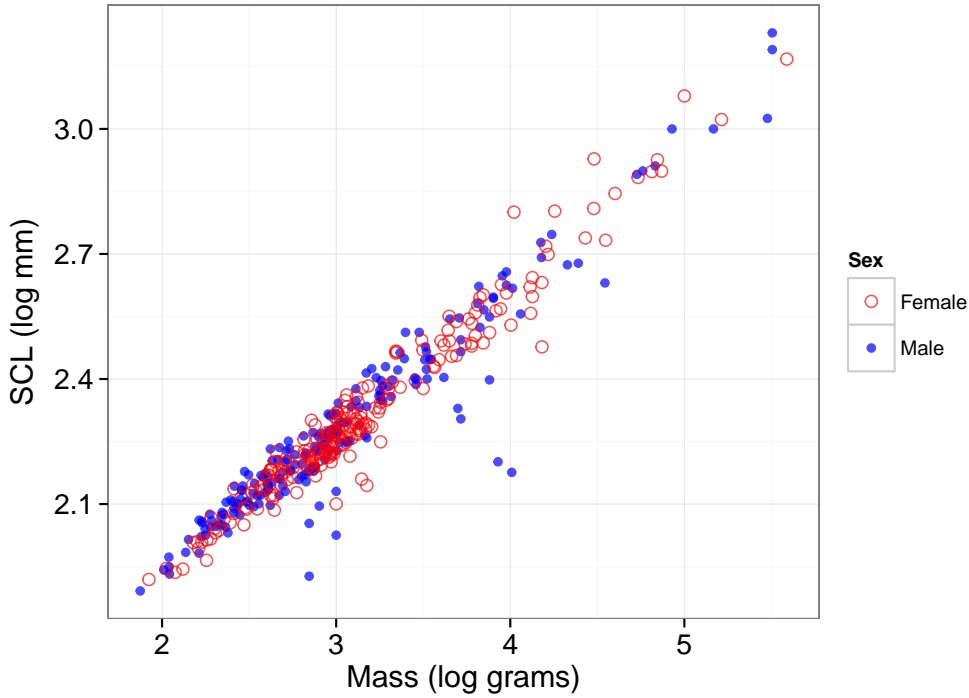

**Figure 3  Regression relationship between body mass and straight carapace length in turtles.** Multiple populations for some species are included. Males are represented by open circles (in blue) and females are closed circles (in red). Males and females did not differ in slope and the difference in $r^2$ was marginally non-significant.

therefore different results partly reflect underlying morphological and scaling differences in addition to differences in statistical power.

The potentially greater statistical power of body mass compared to SCL to detect allometry of SSD emphasizes differences in inferential capabilities that are inherent when using different measures of body size in macroevolutionary studies. These statistical issues likely will be more apparent in clades where SSD is moderate. For example, in our analyses the family Kinosternidae showed moderate female-biased SSD in body mass but moderate male-biased SSD in SCL (Table 1), and showed RR using body mass but isometry with SCL (Table 3). Previous studies using SCL data have found contrasting patterns for RR in Kinosternidae, (e.g., *Berry & Shine, 1980*; *Iverson, 1985*; *Ceballos et al., 2013*; *Halámková, Schulte & Langen, 2013*), with *Ceballos & Iverson (2014)* suggesting that different conclusions have resulted from use of different phylogenetic topologies in comparative analyses. We suggest that choice of body size metric will also influence inferences of RR, particularly in families with moderate or contrasting patterns of SSD. For Kinosternids, males are longer but weigh less than females (at the level of family means), and it is reasonable to assume that similar contrasting patterns of SSD are not uncommon across animals.

Our analyses of SSD in turtles indicate that use of both length and weight metrics would provide more comprehensive evaluation of size dimorphism across lineages, and that larger datasets for each type of metric would be more likely to converge on quantitatively similar

inferences. Although carapace length is the most common body size metric for turtles, we've noted during the course of our data collection that mass is frequently recorded but rarely reported. In numerous instances, authors paradoxically state that they collected mass data for males and females and then never provide or even mention these data again, even in studies of body condition or SSD. However, this problem is not unique to turtles, as available measures of length predominate over mass in non-avian reptiles, generally. While we found that carapace length is likely an acceptable measurement in turtles, we expect that inferences from body size differences using other length measurements in other groups might be inadequate. Turtles are relatively constrained in body shape, whereas squamates are markedly labile. Snout–vent length (SVL) is approximately the length of the torso, but this measurement encapsulates little of the substantial variation in squamate body shape (*Feldman & Meiri, 2013*). Partial limb reduction or limblessness has evolved many times in squamates (*Wiens, Brandley & Reeder, 2006*), tail autotomy is common, and the relationship between SVL (or total body length) and body mass differs dramatically between taxa (*Pincheira-Donoso et al., 2011*).

Body mass has certain advantages over measures of length as a body size metric. Unlike most measures of length, body mass is applicable to all organisms, and estimates do not rely on anatomical landmarks. While both mass and length can be estimated with various levels of precision, length measurements always require at least two anchor points and therefore have at least two components of error variance. Furthermore, body mass is more directly related to physiology, life history, and reproductive biology. The most often cited disadvantage of body mass is that it is purportedly more variable than length measurements owing to differences in body condition and in reproductive and digestive states. We see two problems with this line of reasoning. First, although the statement appeals to intuition because mass effectively compounds three linear dimensions, and therefore a coefficient of variation in mass that is three times higher than that of a single linear measurement would reflect similar precision (*Calder III, 1996*), we know of no studies that have demonstrated that mass measures are more intrinsically variable than length measures at macroevolutionary scales. Most studies that invoke this reasoning are snapshot studies, and their samples do not represent temporal series that would reflect such variation with any type of body size metric anyway. In contrast to this typical argument, our data indicate that more variation in SCL is explained by body mass in females rather than in males. Given the assumption that for turtles it is difficult to assess gravidity reliably in the field, we would expect *a priori* that less variation would be explained by the relationship between SCL and mass in females than in males; therefore, our macro-scale datasets do not support the hypothesis that mass data are intrinsically more variable. Second, we question whether a more variable body size metric, one that reflects short-term fluctuations in body state, would be less informative than a more static body size metric.

Recognizing the many limitations of length datasets, particularly for larger-scale comparisons, several authors have explored the feasibility of interpolating body mass from allometric regressions of length datasets for various vertebrate groups, yielding LDM datasets (e.g., *Pough, 1980*; *Iverson, 1982*; *Pincheira-Donoso et al., 2011*; *Feldman & Meiri, 2013*; *Meiri et al., 2013*; *Feldman et al., 2016*). These equations are based on the relatively

small number of species for which body mass data are available, and as expected, accuracy of estimates will depend partly on the taxonomic scale for which data are generated (most recent interpolations of mass are based on family-specific regression equations). In many instances, vast datasets have been compiled based solely on the interpolations and associated assumptions of relatively few regression equations (e.g., *O'Gorman & Hone, 2012*).

Although the additional data processing involved in generating LDM datasets might provide a suitable heuristic for some purposes, these calculated datasets should not be conflated with actual mass datasets (i.e., species-specific estimates of body mass generated from actual weights). In addition to relying on the assumption that mass and length are perfectly correlated, interpolations could magnify the compounded measurement error associated with length measurements without explaining any more variation. A more pernicious influence is that LDM data could be mistaken for genuine mass measurements by later workers (*Smith & Jungers, 1997*), especially as citations for data points in large-scale analyses are increasingly relegated to unindexed appendices, if at all (*Payne et al., 2012*). Thus, while important inferences have been made from LDM data, we consider results from studies dependent on such datasets to be tenuous, and suggest that results of these studies should be verified with actual mass datasets as they become available. However, we also note that recent advances in computed tomography technology and volumetric modeling, as well as incorporating multiple anatomical correlates of mass, have allowed considerable refinements in weight estimation, particularly in applications to paleontological data (*Field et al., 2013*; *Brassey & Sellers, 2014*; *Brassey et al., 2016*; *Martin, 1990*). For clarity, and because mass and length may point to different conclusions, we recommend that authors analyze the body size measurement that they have, in addition to any desired transformation (e.g., LDM).

While our analyses add to the evidence that the Chelonian clade overall displays the RR pattern of sexual size dimorphism, using either mass or carapace length, the ultimate meaning to be drawn from the allometric pattern remains unclear. The traditional interpretation of RR, that sexual selection for larger males drives body size evolution (*Abouheif & Fairbairin, 1997*) in a (usually male size-biased) group, seems to bear little relevance to most turtle species, which are a predominately female size-biased group. Moreover, the prevailing consensus that SSD in turtles is caused by sex-specific differences in size at sexual maturity is somewhat tautological, and does not indicate ultimate mechanisms involved in these growth differences (*Gibbons & Lovich, 1990*; *Lovich, Gibbons & Agha, 2014*). With an ancestral female size-bias (*Ceballos et al., 2013*) and approximately two-thirds of extant species retaining a female size-bias, ascribing sexual selection for large males as the primary driving force of SSD in Chelonians is an oversimplification. We found the most support (i.e., with and without phylogenetic comparative methods) for RR in two families divergent in their morphologies, habitats, and ecologies: the male size-biased Testudinidae and the female size-biased Trionychidae. It is not immediately apparent how the various overlapping and often contrasting selective forces (e.g., fecundity selection, sexual selection, energetic constraints) would conspire to produce the RR pattern in these families and not in others. Whether the turtle families with an isometric pattern of SSD are constrained by genomic covariation on body size, or display isometry as a result of

 

other forces on male and female size, cannot be answered from examination of RR alone. Rensch's Rule, or his "conjecture" (see *Webb & Freckleton, 2007*), has stimulated much research and discussion of sexual size dimorphism. However, more precise hypotheses and more precise quantification of intra- and interspecific selection forces on body size are needed to understand SSD.

## ACKNOWLEDGEMENTS

We thank all the researchers whose work contributed to our data. Kenneth Angielczyk, Robert Burroughs, and Chris Feldman provided the phylogenetic tree used in our analyses. We thank Ben Anders for assistance with literature and for discussions on body size evolution in turtles. Robert Burroughs, Christian Cox, Kristin Herrmann, and Phil Sudman provided valuable comments on earlier drafts of this manuscript.

### Funding

Koy W. Regis was partly supported by a Tarleton State University Organized Research Grant to JMM. The funders had no role in study design, data collection and analysis, decision to publish, or preparation of the manuscript.

### Grant Disclosures

The following grant information was disclosed by the authors:
Tarleton State University Organized Research Grant.

### Competing Interests

The authors declare there are no competing interests.

### Author Contributions

- Koy W. Regis conceived and designed the experiments, performed the experiments, analyzed the data, wrote the paper, prepared figures and/or tables, reviewed drafts of the paper.
- Jesse M. Meik conceived and designed the experiments, wrote the paper, prepared figures and/or tables, reviewed drafts of the paper.

### Data Availability

    The raw data and R code have been supplied as Supplementary Files.

### Supplemental Information

Supplemental information for this article can be found online at http://dx.doi.org/10.7717/peerj.2914#supplemental-information.

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
