# Peer review of "Allometry of sexual size dimorphism in turtles: a comparison of mass and length data"

_PeerJ, doi:10.7717/peerj.2914_

## Round 0.1 · original submission · Major Revisions

The reviewers were of mixed opinions on the manuscript. All have included valuable comments for the improvement of the manuscript. One of the reviewers (Reviewer 2) had significant concerns regarding the data sources and findings that will need to be directly addressed in any revision. Please explicitly address the editorial changes requested by each of the reviewers in your response.

Reviewer 1 ·

Basic reporting

This paper addresses an important topic in turtle ecology. The problem is well laid out in the introduction. The statistics are complete and understandable, but not overdone. The conclusions are warranted.

My only serious concern is that the authors do not distinguish between the two ways of measuring straight carapace length. Some turtle biologists measure it along the midline within the notches and some measure the maximum size with the caliper bar parallel to the vertebral axis. Rarely do people do it both ways. The latter results in a larger number than the former. This issue is important in the Emydidae and Chelydridae, but not in the Kinosternidae. The authors should address this point and check to see if it alters their results.

Minor things:
line 2 should be Among, not between
153 "for the first time"
255-256 male-female in first half of the sentence, but female-male in the second half. Should be consistent
382 I think it should be Although, not While.
425 third author name problem
466 italics for the scientific name

Experimental design

No comments.

Validity of the findings

No comments.

Additional comments

Well done.

Reviewer 2 ·

Basic reporting

The article is generally well-written and focuses on a topic of interest to biologists. However, the authors appear to have missed some important literature and results from previous studies in their presentation and analyses.

Experimental design

I am concerned about the data used and analyzed for several reasons. First, they do not use the most current list of turtle species/taxonomy. Second, they use questionable data from captive populations (a major concern due to differences in size and growth of captives). Third, they do not exclude all data on juveniles and non-gravid females when possible. Fourth, they didn't list the complete formulae for calculating SSD indices based on Lovich and Gibbons (1992). It is possible that they used data from incorrect calculations. Fifth, the authors are not familiar with the literature questioning the use of mass data and its variation in turtles due to gravidity, gut contents, health, etc. This can confound data seriously.

Validity of the findings

I question the validity of the findings as a result of the issues identified in the experimental design comments above.

Additional comments

Your paper is interesting but the analyses really must be done over again after removing questionable data and ensuring that the analyses are based on the correct formulae. I inserted numerous comments to help you improve the ms should you chose to revise.

Annotated reviews are not available for download in order to protect the identity of reviewers who chose to remain anonymous.

·

Basic reporting

At present the article has a raw data file associated with it. But authors cite the use of multiple R-packages. To accurately reproduce their findings, I request the authors add an additional supplement providing the R-Code used in analyses. This would clarify, as well, the organization of the raw-data file which is difficult to interpret, currently.

Experimental design

No comment

Validity of the findings

No Comments

Additional comments

First, I commend the authors on an excellent an interesting study, one that absolutely addresses a critical need in organismal biology. Below I outline suggested changes based on the line number of the reviewing document:

Lines 112-114. Authors state "few studies have directly compared", but do not provide a citation(s) of some of the few. It would be of use to be able to reference the ones that have compared length and mass measurements.
Lines 114-116 - The authors point out that Rensch combined data for conclusions. Merely a comment to the author's here - based on your results should we reinterpret Rench's findings? I find your conclusions compelling and would implore you to at least discuss this, briefly.
Lines 166-167- Authors say "few instances maximum values were used". What taxa are the "few instances". Given the limited number used here, please be explicit.
Lines 169-171 - Authors say, "For three species" - which species? Please be explicit.
Lines 199-202 - The beginning of this sentence "to account for the non-independence of species" - is unclear to me. After contemplation, I believe the authors are referencing non-independence of mass and CL within closely related species? This is, what they test with Blomberg's K. I recommend they be explicit about what is non-independence. - One could interpret, as I did initially, "non-independence of species", but species are independent evolutionary units! (Except when they are not, of course). Anyways, a clarification here would improve readability.
Lines 235-246 - Authors provide a data summary. Upon review of the author's raw data file, it is still unclear to me which species and populations made it into the final dataset. I request the authors modify the raw data file to indicate inclusion in the final dataset. AND - see suggestion above - they add their R-code to supplement to allow the easiest reproduction of their data and results. - I recognize that this can be an onerous task, but for replicability, it is necessary.
Lines 359-381 - Here the authors discuss a fascinating problem, the potential for deriving mass data from length measurements, in other works. This is a point that the authors make here in the discussion, but I hope they have considered that they have the data available to test precisely this scenario. A comparison with interpolated body-mass against length with their mass/SCL data would allow them to directly address this critical point. I recognize that this is tangentially related to the Rench's Rule focus of their manuscript. If the authors are compelled the addition of this study would be fantastic, but I leave this up to their discretion. I do implore the authors to conduct such a study, independently, if they choose not to include it here.

Final comments, I would like to see a brief discussion by the authors about the application of their methods and data to paleontology. I find their data compelling with respect to the limitations of the fossil record (i.e., we get length data from fossils, not mass). Their discussion grants some leeway to a potentially conservative estimate of SSD based on length. Because it is difficult to sex fossils and get mass estimates, do the authors feel that their methods and data, may be beneficial in this realm? I do. I suggest the authors review some of the literature surrounding mass estimations in fossils, the following citation by Field et al. 2013 on body mass and length measurements applied to birds should prove a useful reflection.

PLoS ONE 8(11): e82000.doi: 10.1371/journal.pone.0082000.

---

## Round 0.2 · accepted · Accept

The reviewers found that you have successfully addressed their concerns, and were appreciative of your rapid turnaround and attention to detail.

Reviewer 1 ·

Basic reporting

“This paper addresses an important topic in turtle ecology. The problem is well laid out in the introduction. The statistics are complete and understandable, but not overdone. The conclusions are warranted.” – no changes in my original conclusions.

The author’s rebuttal was thorough and adequately answered reviewer’s concerns. However, I consider their rebuttal for the comment on use of captive data inadequate. Why did they not just exclude those few sources? They are a source of noise that could have simply been eliminated from the discussion without affecting the results in any way.

Their data sources are not an exhaustive review of the literature containing length and mass data on turtles. One example of a useful resource is Mitchell’s 1994 Reptiles of Virginia, which has statistical data (length and mass) on males and females for 18 species of freshwater turtles, some with large sample sizes.

Experimental design

no comment

Validity of the findings

The conclusions are valid and constitute a valuable contribution to the literature on chelonians and Rensch's Rule.

Additional comments

The paper is well written.

·

Basic reporting

No Comment.

Experimental design

No Comment.

Validity of the findings

No Comment.

Additional comments

I applaud the authors on addressing the concerns of myself and the other reviewers in both a timely and comprehensive manner. Concerning my previous comments regarding clarity of supplemental material and R-Code, the authors have now included R-Code and clarified the supplement to make maximum replication of their efforts possible.

Authors have further addressed my general concerns regarding the conclusions and discussion.

My only additional note is the on the cover page, author J.M. Meik has a superscript 2 associated with their name and no superscript 2 affliation is provided. If there is an additional affiliation this should of course be included.